# New *Dothideomycetes* from Freshwater Habitats in Spain

**DOI:** 10.3390/jof7121102

**Published:** 2021-12-20

**Authors:** Viridiana Magaña-Dueñas, José Francisco Cano-Lira, Alberto Miguel Stchigel

**Affiliations:** Mycology Unit, Medical School, Universitat Rovira i Virgili, C/Sant Llorenç 21, 43201 Reus, Tarragona, Spain; qfbviry@hotmail.com (V.M.-D.); albertomiguel.stchigel@urv.cat (A.M.S.)

**Keywords:** *Dothideomycetes*, freshwater fungi, taxonomy, phylogeny

## Abstract

The *Dothideomycetes* are a class of cosmopolitan fungi that are present principally in terrestrial environments, but which have also been found in freshwater and marine habitats. In the present study, more than a hundred samples of plant debris were collected from various freshwater locations in Spain. Its incubation in wet chambers allowed us to detect and to isolate in pure culture numerous fungi producing asexual reproductive fruiting bodies (conidiomata). Thanks to a morphological comparison and to a phylogenetic analysis that combined the internal transcribed spacer (ITS) region of the nrDNA with fragments of the RNA polymerase II subunit 2 (*rpb*2), beta tubulin (*tub*2), and the translation elongation factor 1-alpha (*tef-*1) genes, six of those strains were identified as new species to science. Three belong to the family *Didymellaceae: Didymella brevipilosa, Heterophoma polypusiformis* and *Paraboeremia clausa*; and three belong to the family *Phaeosphaeriaceae:*
*Paraphoma aquatica, Phaeosphaeria fructigena* and *Xenophoma microspora*. The finding of these new taxa significantly increases the number of the coelomycetous fungi that have been described from freshwater habitats.

## 1. Introduction

Freshwater fungi are a taxonomically heterogeneous ecological group of organisms of cosmopolitan distribution playing an important ecological role in the recycling of dead organic matter [1], where some of them are restricted to tropical or temperate areas, and others are present in cold-water habitats [2] Most freshwater fungi belong to the phylum Ascomycota (the ascomycetes), whose biodiversity depends on their geographical location and substrates [3]. Around 740 species of ascomycetes have been reported in freshwater habitats (http://fungi.life.illinois.edu/search/world_records, accessed on 10 November 2021), and approximately one-third of these belong to the class *Dothideomycetes* [4,5].

The *Dothideomycetes* is among the earliest fungi that were reported in freshwater environments [5]. They are a group of fungi characterized by the production of fissitunicate (bitunicate) asci in unilocular and polylocular ascomata [6,7], currently classified into two subclasses: the *Dothideomycetidae*, comprising the orders *Capnodiales, Dothideales,* and *Myriangiales;* and the *Pleosporomycetidae*, comprising the orders *Gloniales, Hysteriales, Jahnulales, Mytilinidiales,* and *Pleosporales* [8,9]. The majority of freshwater *Dothideomycetes* belong to the *Pleosporales* and to the *Jahnulales* [5].

The *Pleosporales* is the largest order of the *Dothideomycetes*, comprising a quarter of its species [6,10]. Taxa in this order have been found in diverse habitats, and can act as saprobes, endophytes, pathogens, or parasites. Most of the *Pleosporales* are plant pathogens with a wide range of hosts and mainly cause leaf and stem lesions [6,9,11,12].

The diverse hydrography, topology, and climatology of Spain led to the formation of several well-defined ecological regions that have a wide spectrum of scarcely explored habitats potentially rich in fungal populations. During this study, we isolated several fungi living on plant debris in freshwater habitats. The objective was to perform phylogenetic analyses using the nucleotide sequences of informative molecular markers that can clarify the taxonomy of these fungal isolates, and to describe any noteworthy taxa.

## 2. Materials and Methods

### 2.1. Samples Collection and Fungal Isolation

A total of 119 samples of plant debris submerged in freshwater in Spain were collected as follows: two in *Parque de Doña Casilda Iturriza* (Vizcaya province), three in *Les Guilleries* (Barcelona province), ten in *Cúber* (Escorca, Mallorca), 22 in *Capafonts* (Tarragona province), 15 in *Pontons* (Barcelona province), 17 in *Riaza* (Segovia province), and 50 in *Cascadas del Huéznar* (Cazalla de la Sierra, Sevilla province). Samples were placed into sterile self-sealing plastic bags for transport to the laboratory and stored at 5 °C. Samples were processed and examined following the method described previously by Magaña-Dueñas et al. [13]. All isolates were stored in the culture collection at the Faculty of Medicine and Health Sciences (FMR), Reus (Spain), and herborized materials and living cultures of novel fungi were deposited in the fungal collection at Westerdijk Fungal Biodiversity Institute (CBS; Utrecht, The Netherlands). The nomenclature and descriptions were registered in MycoBank (https://www.mycobank.org/page/Registration%20home, accessed on 4 October 2021).

### 2.2. Phenotypic Study

The phenotypic studies were carried out according to Magaña-Dueñas et al. [14], except for strain FMR 17808, in which the measurements of the structures were carried out after 30 days, due to the slow production of fertile fruiting bodies.

### 2.3. DNA Extraction, Amplification, and Sequencing

DNA extraction, amplification, and sequencing were carried out following the protocols outlined by Magaña-Dueñas et al. [13,14]. SeqMan software v. 7.0.0 (DNAStar Lasergene, Madison, WI, USA) was used to obtain and edit the consensus sequences. Sequences generated in this study were deposited at the European Nucleotide Archive (ENA) (Appendix A).

### 2.4. Phylogenetic Analysis

The sequences obtained were compared with other fungal sequences deposited at the National Center for Biotechnology Information (NCBI) database using the Basic Local Alignment Search Tool (BLAST; https://blast.ncbi.nlm.nih.gov/Blast.cgi, accessed on 20 July 2021). For the phylogenetic study, an alignment for each locus was made using the MEGA (Molecular Evolutionary Genetics analysis) program v. 7.0 [15], using the Clustal W algorithm [16] and refined with MUSCLE [17], or manually, when necessary, on the same platform. Phylogenetic analyses were made by maximum-likelihood (ML) and Bayesian interference (BI) with RAxML v. 8.2.12 [18] software on the online Cipres Science gateway portal [19] and MrBayes v.3.2.6 [20], respectively. The final matrix used for phylogenetic analyses were deposited in TreeBASE (http://purl.org/phylo/treebase/phylows/study/TB2:S295050 (accessed on 25 November 2021)).

For the *Didymellaceae*, the phylogenetic reconstructions were performed using the concatenated nucleotide sequences of three phylogenetic markers (ITS, *rpb2,* and *tub2*), while for the *Phaeosphaeriaceae* four phylogenetic markers (ITS, *rpb2, tub2,* and *tef-1*) were used. The best nucleotide substitution model for the BI analysis of the family *Didymellaceae* was the Kimura 2-parametrer with proportion of Invariable sites and Gamma distribution (K80 + I + G) for *rpb2*, and the Symmetrical model with proportion of Invariable sites and Gamma distribution (SYM + I + G) for ITS and *tub2*. For the *Phaeosphaeriaceae*, the best model was General Time Reversible with proportion of Invariable sites and Gamma distribution (GTR + I + G) for ITS, K80 + I+G for *rpb2*, Hasegawa–Kishino–Yano with Gamma distribution (HKY + G) for *tub2* and General Time Reversible with Gamma distribution (GTR + G) for *tef-1*, all estimated using the program jModelTest [21]. The parameter settings used in BI analyses were two simultaneous runs of 5,000,000 generations, and Markov chain Monte Carlo (MCMC), with samples taken every 1000 generations. The 50% majority rule consensus tree and posterior probability values (PP) were calculated after discarding the first 25% of the resulting trees. A PP value ≥ 0.95 was considered as significant [22]. For ML analysis, support for internal branches was assessed by 1000 ML bootstrapped pseudo replicates. Bootstrap support value (BS) ≥ 70% was considered significant.

## 3. Results

### 3.1. Phylogeny

For the Didymellaceae, the alignment comprised 31 ingroups of strains with a total of 1389 characters, including gaps (476 bp for ITS, 301 for *tub2,* and 612 *rpb2*), which included 493 bp variable sites (111 for ITS, 114 for *tub2,* and 268 for *rpb2*) and 399 bp phylogenetically informative sites (91 for ITS, 81 for *tub2,* and 227 for *rpb2*). *Vacuiphoma oculihominis* UTHSC: DI16-308 and *V. bulgarica* CBS 357.84 were used as outgroup. The BI analysis showed similar tree topology and was congruent with that obtained in the ML analysis. For the BI multilocus analysis, a total of 1352 trees were sampled after the burn-in with a stop value of 0.01. In the phylogenetic tree (Figure 1), our strains were placed in three well-supported clades. The *Paraboeremia* clade (100% BS/1 PP) included all the described species of the genus together with two of our strains (FMR 18598 and FMR 18597), which were placed in a fully supported independent terminal branch. The *Didymella* clade (95% BS/1 PP) comprised the seven previously described species of the genus *Didymella* (including the type species *D. exigua*) together with our strain FMR 17415, distant from the rest in an independent terminal branch. The *Heterophoma* clade (100% BS/1 PP) included all six previously described species together with our strain FMR 17837, and was placed in the same terminal clade as *H. verbasci-densiflori*.

For the *Phaeosphaeriaceae*, the alignment included 32 ingroups of strains, with a total of 2256 characters including gaps (422 bp for ITS, 699 for *rpb2*, 282 for *tub2,* and 853 for *tef-1*), which comprised 715 bp variable sites (161 for ITS, 288 for *rpb2*, 94 for *tub2,* and 172 for *tef-1*) and 507 bp phylogenetically informative sites (130 for ITS, 72 for *tub2*, 219 for *rpb2,* and 86 for *tef-1*). *Neophaeosphaeria filamentosa* CBS 102202 and *N. agaves* CBS 136429 were included as outgroup. For the BI multi-locus analysis, 1378 trees were sampled after the burn-in with a stop value of 0.01. The phylogenetic tree included nine genera of the *Phaeosphaeriaceae* (Figure 2). To resolve the phylogenetic placement of our strain FMR 17808, a phylogenetic tree was constructed using the LSU sequences of *Phaeosphaeria*, since for most species of the genus the sequences of other markers (*rpb2, tef-1*, and ITS) are not available (Appendix A). In the *Phaeosphaeriaceae* tree (Figure 2), the *Phaeosphaeria* clade (91% BS/1 PP) included eight previously described species together with FMR 17808, which was placed in a terminal independent branch distant from the rest of the species. The *Xenophoma* clade (100% BS/1 PP) included the type species *X. puncteliae* together with our strain FMR 17947. The *Paraphoma* clade (70% BS/0.90 PP) comprised all the described species of the genus and our strain FMR 16956, which was placed in a full supported terminal branch together with *P. radicina.*

### 3.2. Taxonomy

*Didymellaceae* Gruyter et al., Mycol. Res. 113: 516. 2009.


*Type genus: Didymella Sacc.*


*Basionym*: *Peyronellaea* Goid. ex Togliani, Ann. Sperim. Agrar. II 6: 93. 1952.

*Didymella* Sacc. ex Sacc., Syll. Fung. 1: 545. 1882. Emend. Chen et al., Stud. Mycol. 82: 173. 2015.

*Type species: Didymella exigua* (Niessl) Sacc.

*Didymella brevipilosa* V. Magaña-Dueñas, Stchigel and Cano, sp. nov. MycoBank MB841361 (Figure 3).

*Etymology*. From Latin *brevivus*-, short, and -*pilosae*, hairy, because the short setae surrounded the neck.

*Type:* Spain, Segovia province, *Riaza* (41.238863, −3.435258), from plant debris submerged in freshwater, May. 2018, col. Viridiana Magaña Dueñas, holotype CBS H-24906. living cultures FMR 17415 = CBS 148654.

*Description:* Hyphae hyaline to pale brown, septate, branched, smooth- and thin-walled, 1.5–2.5 µm wide. Conidiomata pycnidial, brown to dark brown, immersed to semi-immersed, solitary, scattered, setose, ostiolate, mostly subglobose, 200–400 × 160–410 µm; 1 to 3 ostiolar necks, 50–60 × 60–75 µm, ostiole 50–60 µm diam. Setae subhyaline to brown, septate, slightly sinuous, nodose and verrucose, 15–100 µm, tapering towards the apex, which is slightly apiculate, mostly arranged around the neck. Conidiomata wall 4–6-layered, 15–30 µm thick, with an outer layer of *textura angularis*, composed of light brown to dark, flattened polygonal cells of 5–8 µm diam. Conidiophores absent. Conidiogenous cells phialidic, determinate, hyaline, smooth-walled, ampulliform to globose, 5–8 × 4–6 µm. Conidia aseptate, hyaline, smooth- and thin-walled, bacilliform to kidney-shaped, 4–5 × 2–3 µm. Chlamydospores absent.

*Culture characteristics*: Colonies on PDA reaching 38–41 mm diam after 7 days at 25 ± 1 °C, flattened, slightly floccose, margin undulate, olive brown (4F7) with white patches, border yellowish grey (4B2); reverse brownish grey to greyish yellow (8F2/4B3). Colonies on OA reaching 40–43 mm diam after 7 days at 25 ± 1 °C, flattened, granular due to abundant pycnidia, margin regular, surface and reverse grey (5F1). Colonies on MEA reaching 37–39 mm diam after 7 days at 25 ± 1 °C, flattened, velvety, margins undulate, dark brown to greyish yellow (6F7/4B4), border yellowish grey (4B2); reverse dark brown to greyish orange (6F7/4B4) border greyish yellow (4B4). Exopigment absent. Cardinal temperatures for growing—optimum 25 °C, maximum 30 °C, minimum 5 °C.

*Diagnosis*: Morphologically, *Didymella brevipilosa* differs from the rest of the species located in the same clade in having slightly sinuous, nodose and verrucose setae mainly on the neck and around the pycnidial ostiole, while the other species produce glabrous conidiomata [23,24]. In the submerged plant material, from which the fungus was isolated, only the asexual stage was observed.

*Notes*: Regarding the ITS-*rpb2*-*tub2* concatenated sequences alignment, the nucleotide differences between *D. brevipilosa* and the other species in the same terminal clade were: *D. macrostoma,* 110 bp; *D. pteridis,* 120 bp; *D. subrosea,* 97 bp; and *D. viburnicola,* 101 bp.

*Heterophoma* Qian Chen and L. Cai, Stud. Mycol. 82: 165. 2015.

*Type species: Heterophoma sylvatica* (Sacc) Qian Chen and L. Cai.

Because the sexual stage of *Heterophoma* has not been previously reported and described, we emend the generic description as next:

*Description:* Sexual stage: Ascomata superficial, solitary, non-ostiolate, dark brown, opaque, lens shaped to subglobose; *hamathecium* comprising numerous hyaline, septate, filamentous paraphyses; asci 8-spored, cylindrical, with an apical annular apparatus; ascospores two-celled, hyaline and biconic when young, becoming muriform and brown to dark brown at maturity, with 5 transversal septa (frequently developing 1–2 additional transverse septa) and 2–4 longitudinal and oblique septa, broadly fusiform, frequently constricted at the middle septum, occasionally constricted at other septa, surrounded by a mucilaginous sheath. Asexual stage: Conidiomata pycnidial, globose to subglobose, superficial on or immersed into the agar, solitary or confluent, ostiolate; pycnidial wall pseudoparenchymatous, 5–12-layered; conidiogenous cells phialidic, hyaline, smooth, ampulliform to doliiform; conidia variable in shape and size, hyaline, smooth- and thin-walled, 0–1(–2)-septate, i.e., ellipsoidal, oblong, cylindrical, reniform, or slightly allantoid, mostly guttulate. Chlamydospores unicellular, globose, intercalary in chains, olivaceous.

*Heterophoma polypusiformis* V. Magaña-Dueñas, Cano and Stchigel, sp. nov. MycoBank MB841362 (Figure 4).

*Etymology*. From Latin *polypus*-, octopus, -*formis*, shape, because the morphological resemblance of immersed conidiomata to an octopus.

*Type*: Spain, Sevilla province, *Cascadas del Huéznar* (37.9935997, −5.6718387), from plant debris in freshwater, May. 2019, José F. Cano Lira, holotype CBS H-24907, living cultures FMR 17837 = CBS 148655.

*Description:* Hyphae hyaline, septate, branched, smooth- and thin-walled, 2–3 µm wide. Sexual stage: Ascomata superficial, solitary, non-ostiolate, dark brown, opaque, lens shaped to subglobose, up to 600 × 400 µm, thin-walled; hamathecium comprising numerous hyaline, septate, filamentous paraphyses, 1–3 µm wide; asci 8-spored, cylindrical, 90–100 × 10–15 µm, with an apical, annular apparatus; ascospores two-celled, hyaline and bi-conic when young, becoming muriform and brown to dark brown at maturity, with 5 transversal septa (frequently developing 1–2 additional transverse septa) and 2–4 longitudinal and oblique septa, broadly fusiform, 10–35 × 5–10 µm, frequently constricted at the middle septum, occasionally constricted at other septa, surrounded by a mucilaginous sheath. Asexual stage: Conidiomata pycnidial, semi-immersed, brown to dark brown, solitary, scattered, ostiolate, setose, subglobose, 160–180 × 170–200 µm, developing one to a few necks turning paler towards the ostiole, cylindrical 60–150 µm long, ostiole 60–85 µm diam; when the conidiomata grow immersed in the medium (OA) develop numerous necks that branch out, giving them a cephalopod look to the pycnidia; setae hyaline to sub-hyaline, septate, nodose, thick-walled, 15–40 µm long, mainly disposed around the ostiole, rounded and curved at the tip, conidiomata wall 4–6-layered, 10–25 µm thick, with an outer layer of *textura angularis* composed of brown to dark brown, flattened polygonal cells of 4–8 µm diam; conidiogenous cells phialidic, determinate, hyaline, smooth-walled, doliiform, 4–6 × 3–4 µm; conidia aseptate, hyaline, smooth- and thin-walled, cylindrical, 4–5 × 1.5–2 µm. Chlamydospores aseptate, intercalary, smooth-walled, brown, globose, 11–15 µm diam.

*Culture characteristics*: Colonies on PDA reaching 59–60 mm diam after 7 days at 25 + 1 °C, flattened, velvety, margin regular, orange grey to yellowish (6B2/5D4); reverse dark brown (6F7), margins orange white (6A2). Colonies on OA reaching 39–42 mm diam after 7 days at 25 + 1 °C, flattened, slightly cottony, margin regular, surface and reverse orange white (6A2). Colonies on MEA reaching 43–45 mm diam after 7 days at 25 + 1 °C, flattened, velvety, margins undulate, white to orange white (6A2); reverse greyish orange (5B3). Exopigment absent. Cardinal temperatures for growing—optimum 25 °C, maximum 35 °C, minimum 5 °C.

*Diagnosis*: *Heterophoma polypusiformis* is easily distinguishable from the other species of the genus because it produces a sexual stage, which is morphologically related to the genus *Ascochyta*, a member of the *Didymellaceae* [25]. Moreover, *H. polypusiformis* develops into OA pycnidia with numerous ostiolar necks that branch out giving them a cephalopod appearance, a feature never seen in the other species of the genus. Unlike *H. polypusiformis,* its phylogenetically closer species, *H. verbasci-densiflori*, produces in the same culture conditions pycnidia bearing 1-6 papillated to short (of less than 60 µm long, whereas these can reach up to 200 µm in *H. polypusiformis*) unbranched necks (branched in *H. polypusiformis*), with a ‘potato-like’ appearance. The sexual stage of *H. polypusiformis* was found on the submerged plant material, whereas its asexual stage was observed once the fungus was grown in pure culture.

*Notes*: Differences between the nucleotide sequences (ITS-*rpb2*-*tub2*concatened dataset) of *H. polypusiformis* and *H. verbasci-densiflori* were of 16 bp.

*Paraboeremia* Q. Chen and L. Cai, Stud. Mycol. 82: 183. 2015.

*Type species: Paraboeremia selaginellae* (Sacc.) Q. Chen & L. Cai.

*Paraboeremia clausa* V. Magaña-Dueñas, Stchigel and Cano, sp. nov. MycoBank MB841363 (Figure 5).

*Etymology*. From Latin *clausa*, closed, because the absence of conidiomata ostioles.

*Type*: Spain, Vizcaya province, Bilbao, *Parque de Doña Casilda Iturriza* (43.2658246, −2.942885), from freshwater submerged plant debris, Aug 2020, Viridiana Magaña Dueñas, holotype CBS H-24908, living cultures FMR 18597 = CBS 148656.

*Other material examined:* Spain, Vizcaya province, Bilbao, *Parque de Doña Casilda Iturriza* (43.2658246, −2.942885), from freshwater submerged plant debris, Aug. 2020, Viridiana Magaña Dueñas, living cultures FMR 18598.

*Description:* Hyphae pale brown to brown, septate, branched and smooth-walled, 2.5–5 µm wide. Conidiomata pycnidial, sub-hyaline to pale brown, translucent, immersed to semi-immersed, solitary, scattered, barrel-shaped to pyriform, 270–480 × 270–300 µm, covered by brown, septate, smooth to asperulate, thin-walled anastomosing hyphae; neck absent or rarely present, conical-truncate, 120–130 × 180–200 µm, ostiole indistinguishable. Conidiomata wall 4–6-layered, 25–35 µm thick, with an outer layer of *textura intricata*, composed of pale brown to brown hyphae of 2–4 µm wide. Conidiophores absent. Conidiogenous cells phialidic, determinate, hyaline, smooth-walled, globose, 5.5–6.5 × 5.5–7 µm. Conidia aseptate, hyaline, smooth- and thin-walled, broadly ellipsoidal to ovoid, 3–3.5 × 1.5–2.5 µm, one- or biguttulate. Chlamydospores absent.

*Culture characteristics*: Colonies on PDA reaching 37–40 mm diam after 7 days at 25 + 1 °C, flattened, velvety, margin regular, yellowish brown (5F4) with a yellowish white (4A2) border; reverse with the same colour than the surface. Colonies on OA reaching 55–57 mm diam after 7 days at 25 + 1 °C, flattened, granular due to abundant production of pycnidia, margin filamentous, brown (6F4), reverse greyish brown (6F3). Colonies on MEA reaching 41–44 mm diam after 7 days at 25 + 1 °C, flattened, radiate, velvety, margins regular, dark brown to greyish brown to (8F8/8F3), border orange white (6A2); reverse brownish grey to greyish brown (8F2/7E3) border orange white (6A2). Exopigment light orange (5A4). Cardinal temperatures for growing—optimum 25 °C, maximum 30 °C, minimum 5 °C.

*Diagnosis*: *Paraboeremia clausa* is phylogenetic close to *P. putaminum*, but differs from the latter because the pycnidia lack ostioles, and the conidia are hyaline in mass, while those of *P. putaminum* are greenish [26]. Both strains of *P. clausa* displayed similar phenotypic features in pure culture, and the asexual stage of *P. clausa* was originally detected on the freshwater submerged plant debris.

*Notes*: Differences between ITS-*tub2*-*rpb2* concatenated nucleotide sequences of *P. clausa* and *P. putaminum* were 15 bp.

*Phaeosphaeriaceae* M. E. Barr Mycologia 71(5): 948. 1979.

*Type genus: Phaeosphaeria* I. Miyake.

*Paraphoma* Morgan-Jones and J.F. White, Mycotaxon 18 (1): 58. 1983.

*Type species*: *Paraphoma radicina* (McAlpine) Morgan-Jones and J.F. White, Mycotaxon 18 (1): 60. 1983.

*Paraphoma aquatica* V. Magaña-Dueñas, Stchigel and Cano, sp. nov. MycoBank MB841364 (Figure 6).

*Etymology*. From Latin *aquaticus*, referring to the habitat from which the fungus was recovered.

*Type*: Spain, Barcelona province, *Les Guilleries* (41.9362028, 2.4122862), from freshwater submerged plant debris, Nov. 2017, Eduardo Jose de Carvalho Reis, holotype CBS H-24909, living cultures FMR 16956 = CBS 148657.

*Description:* Hyphae pale brown to brown, septate, branched, smooth- and thin-walled, 1.5–2.5 µm wide. Conidiomata pycnidial, dark brown, semi-immersed, solitary, scattered, setose, globose to subglobose 380–570 × 400–570 µm, non-ostiolate. Setae brown, septate, smooth- and thick-walled, rounded at the tip, 90–150 µm. Conidiomata wall 5–7-layered, 15–30 µm thick, with an outer layer of *textura angularis*, composed of brown to dark brown, flattened polygonal cells of 5–8 µm diam, covered by a mass of interwoven, brown hyphae. Conidiophores absent. Conidiogenous cells phialidic, determinate, hyaline, smooth-walled, ampulliform to globose, 4–6 × 5–8 µm. Conidia aseptate, hyaline, smooth- and thin-walled, ellipsoidal, 5–8 × 2–3 µm. Chlamydospores absent.

*Culture characteristics*: Colonies on PDA reaching 54–55 mm diam after 7 days at 25 + 1 °C, flattened, velvety, margin regular, greyish brown to brownish grey (5D3/5F2); reverse yellowish brown to orange grey (5D5/5B2). Colonies on OA reaching 56–59 mm diam after 7 days at 25 + 1 °C, umbilicate, velvety, margin regular, dull green (30E4); reverse greenish grey to dull green (30F2/30E4), border white. Colonies on MEA reaching 45–48 mm diam after 7 days at 25 + 1 °C, flattened, velvety, margins regular, olive grey to greenish grey (3D2/3B4), reverse grey to greyish yellow (3F1/3B4), border yellowish white (3A2). Exopigment absent. Cardinal temperatures for growing—optimum 25 °C, maximum 30 °C, minimum 5 °C.

*Diagnosis*: *Paraphoma aquatica* differs from the phylogenetically closest species, *P. radicina*, because its pycnidia lack of a neck and are non-ostiolate. The asexual stage of *P. aquatica* was also observed in the submerged substrate.

*Notes*: *Paraphoma aquatica* is located in the clade with a weak support; however, it forms a fully supported clade with *Paraphoma radicina*. The concatenated ITS-*rpb2*-*tub2*-*tef-1* nucleotide sequences of both species differs in 59 bp.

*Phaeosphaeria* I. Miyake, Bot. Mag., Tokyo 23: 93. 1909.

*Type species*: *Phaeosphaeria oryzae* I. Miyake, Bot. Mag., Tokyo 23: 93. 1909.

*Phaeosphaeria fructigena* V. Magaña-Dueñas, Cano and Stchigel, sp. nov. MycoBank MB841365 (Figure 7).

*Etymology*. From Latin *fructi*-, fruits, and -*genes*, because of the production of ascomata in vitro.

*Type*: Spain, Tarragona province, *Capafonts* (41.29598, 1.02753), from freshwater submerged plant debris, Mar. 2019, Viridiana Magaña Dueñas and Isabel Iturrieta González, Holotype CBS H-24910, living cultures FMR 17808 = CBS 148658.

*Description:* Mycelium superficial to immersed, composed by septate, hyaline, smooth- and thin-walled, branched hyphae, 2–3 μm wide. Ascomata perithecial, immersed to semi-immersed, solitary, ostiolate, with up to three necks, reddish-brown to dark brown, becoming paler towards the top of the neck, pyriform, globose to irregularly-shaped, 230–370 × 210–330 μm; neck conic-truncate, 70–145 × 40–80 μm, ostiole 30–70 μm diam.; peridial wall 2–4-layered, 35–60 μm thick, outer wall of *textura intricata*, composed of brown to dark brown hyphae of 3–4 μm wide. *Hamathecium* comprising numerous, filamentous, septate, branched paraphyses, and pseudoparaphyses of 1.5–2 μm wide. Asci 6–8-spored, bitunicate, cylindrical to cylindrical-clavate, 80–120 × 10–12 μm, without apical structures. Ascospores hyaline when young, becoming pale brown at maturity, three-septate, fusiform, 22–28 × 4–5 μm, narrowly rounded at the ends.

*Culture characteristics*: Colonies on PDA reaching 40–41 mm diam after 7 days at 25 + 1 °C, umbonate, velvety, margin undulate, with abundant aerial mycelium, surface white (6A1), border orange white (6A2); reverse pale brown (6D4), border orange white (6A2). Colonies on OA reaching 46–48 mm diam, flattened to slightly floccose, margins regular, with sparse aerial mycelium, surface and reverse yellowish grey (4B2). Colonies on MEA reaching 39–40 mm diam, flattened, velvety, margin regular, with abundant aerial mycelium, orange grey (5B2); reverse orange white (5A2). Cardinal temperature for growing—optimum 25 °C, maximum 30 °C, minimum 5 °C.

*Diagnosis*: Morphologically, *Phaeosphaeria fructigena* is characterized by the production in vitro of ascomata with up to three necks, and of fusiform ascospores. Only the sexual stage of *P. fructigena* has been observed in both original material and pure culture.

*Notes*: In our phylogenetic analysis *P. fructigena* is located at an independent branch, thus revealing itself as a new species.

*Xenophoma* Crous & Trakunyingcharoen, IMA fungus. 5(2): 404. 2014.

*Type species*: *Xenophoma pucteliae* Crous & Trakunyingcharoen, IMA fungus. 5(2): 404. 2014.

*Basionym*: *Phoma puncteliae* Diederich & Lawrey, Fungal Div. 55: 207 (2013).

*Xenophoma microspora* V. Magaña-Dueñas, Stchigel and Cano, sp. nov. FMR 17947. MycoBank MB841366 (Figure 8).

*Etymology*. From Greek *μικρο*-, small, -*σπόριο*, spore, due to the small size of the conidia.

*Type*: Spain, Barcelona province, *Pontons* (41.41397, 1.52678), from freshwater submerged plant debris, Jun. 2018, Viridiana Magaña Dueñas, Holotype CBS H-24911 living cultures FMR 17947 = CBS 148659.

*Description:* Hyphae hyaline to subhyaline, septate, smooth- and thin-walled, branched, 2.5–3 µm wide. Conidiomata pycnidial, brown to dark brown, immersed to semi-immersed, solitary, scattered, ostiolate, globose to subglobose, 160–180 × 200–250 µm, with up to 3 conic-truncate ostiolar necks of 30–40 × 35–45 µm. Conidiomata wall 4–6-layered, 15–25 µm thick, with an outer layer of *textura angularis*, composed of light brown to brown, flattened polygonal cells of 4–6 µm diam. Conidiophores absent. Conidiogenous cells phialidic, determinate, hyaline, smooth-walled, ampulliform to globose, 2.5–4 × 3–3.5 µm. Conidia aseptate, hyaline, smooth- and thin-walled, bacilliform, 1.5–2.5 × 1–1.2 µm Chlamydospores absent.

*Culture characteristics*: Colonies on PDA reaching 59–60 mm diam after 7 days at 25 + 1 °C, flattened, floccose, margin irregular, surface and reverse olive brown (4D4). Colonies on OA reaching 49–51 mm diam after 7 days at 25 + 1 °C, flattened, velvety, margin regular, surface and reverse yellowish grey with patches olive brown (4B2/4F3). Colonies on MEA reaching 42–44 mm diam after 7 days at 25 + 1 °C, flattened, velvety, margins lobate, olive brown (4F4); reverse brownish grey (4F2). Cardinal temperatures for growing—optimum 25 °C, maximum 30 °C, minimum 5 °C.

*Diagnosis*: Morphologically, *Xenophoma microspora* is distinguished from *Xenophoma puncteliae* by the production of up to three ostiolar necks, and also by producing smaller conidia than *X. puncteliae* (1.5–2.5 × 1–1.2 µm vs. 2.5–3 × 2–2.5 µm). The asexual stage of *X. microspora* was observed in plant debris submerged in freshwater.

*Notes*: Among the concatenated sequences (ITS-*rpb2-tub2-tef-1*), the difference in nucleotides between *X. punctileae* and *X. microspora* is 27 bp.

## 4. Discussion

Recently, molecular biology helped to clarify the phylogenetic relationships between the members of the *Dothideomycetes*, especially among several phoma-like fungal taxa. Multilocus analyses based on LSU, ITS, *rpb2, tef-1,* and *tub2* sequences have been widely used to define the species boundaries for the *Didymellaceae*, the *Phaeosphaeriaceae* and other families of the *Dothideomycetes* [23,24,27,28,29,30]. However, *rpb2* alone provides a phylogenetic tree with a similar topology to those obtained with more phylogenetic markers [24,30,31].

During the development of the present study, we isolated several fungi from submerged wood in certain freshwater habitats of Spain. We carried out phylogenetic analyses with concatenated of three loci (ITS-*rpb2-tub2*) for the *Didymellaceae* members, and of four loci (ITS-*rpb2- tub2-tef-1*) for those taxa in the *Phaeosphaeriaceae*. Thus, we report six new species to science: *Didymella brevipilosa*, *Heterophoma polypusiformis, Paraboeremia clausa, Paraphoma aquatica, Phaeosphaeria fructigena,* and *Xenophoma microspora*.

The species of *Didymella*—genus established by Saccardo [32] to accommodate *D. exigua*—are saprobes commonly found in living and dead parts of herbaceous and woody plants but are also important phytopathogens. Several species have also been isolated from inorganic substrates, such as asbestos, cement, and paint [23,28,33]. In 2015, Chen et al. carried out a multilocus phylogenetic analysis and the genus was defined as monophyletic and encompassing 37 species [28]. Approximately 30 new species have recently been included in the genus [23,24,29,30,34,35,36]. In our phylogenetic analysis, *Didymella brevipilosa* was placed in an independent branch separated from the rest of *Didymella* spp. In addition, this species is characterized by having short, sinuous and asperulate setae mainly located around the ostioles, unlike most of the species of the genus, which lack these structures.

The genus *Heterophoma* was introduced by Chen et al. [28] to accommodate *H. adonidis, H. nobilis, H. novae-verbascicola, H. poolensis,* and *H. sylvatica*. Seven species are currently recognized (http://www.indexfungorum.org/names/Names.asp, accessed on 25 October 2021). Species of this genus are saprobes and plant pathogens (especially on members of the families *Brassicaceae* and *Scrophulariaceae*) with a cosmopolitan distribution [23,28,30]. In our study, we report the finding of *Heterophoma polypusiformis*, the first species of the genus isolated from wood submerged in freshwater. Moreover, *H. polypusiformis* produces both asexual and sexual stages, being the first species of the genus reported to have sexual reproduction. The main features of the ascospores (smooth-walled, muriform, brown, and surrounded by a gelatinous sheath) correspond to those reported for other genera of the family *Didymellaceae*, such as *Ascochyta* and *Neomicrosphaeropsis* [25]. *Heterophoma polypusiformis* is easily distinguishable from the other species of the genus because the pycnidia submerged in the culture medium have an ‘octopus’ appearance.

Chen et al. [28] introduced the genus *Paraboeremia* into the *Didymellaceae* to accommodate *P. adianticola, P. putaminum,* and *P. selaginellae*. Nine species are currently accepted in the genus (http://www.indexfungorum.org/Names/Names.asp, accessed on 25 October 2021). The majority of the species of the genus are plant parasites, causing leaf or stem spots [28,31]. Moreover, *Paraboeremia* spp. have been isolated from the rhizosphere, soil, and healthy and dead plants [23,28,30,31]. *Paraboeremia clausa* is the first species reported in plant material submerged in freshwater and is characterized by the production of barrel-shaped to pyriform, translucent, very pale colored pycnidia covered by dark brown anastomosing hyphae and lacking ostioles.

The genus *Phaeosphaeria* was introduced by Miyake [37] to accommodate its type species, *P. oryzae*. Later, the lectotype was designated by Eriksson [38], and due to the morphological similarities with the *Leptosphaeria*, both genera were for a long time considered synonyms. Barr [39] subsequently established a new family *Phaeosphaeriaceae,* designating *Phaeosphaeria* as its type genus. *Phaeosphaeria* species have a cosmopolitan distribution, they are saprobic but also pathogenic stems, flowers, and leaves of monocotyledons, and hyperparasites of other fungi [27]. There are 219 species currently listed in the Index Fungorum (http://www.indexfungorum.org/Names/Names.asp, accessed on 25 October 2021). *Phaeosphaeria fructigena* was isolated from plant debris submerged in freshwater, and its sexual stage shares several features with other species of the genus (e.g., *P. musae, P. oryzae,* and *P. thysanolaenicola*), such as the fissitunicate asci and 3-septate, hyaline to pale yellowish ascospores [27,40]. In our phylogenetic analysis *P. fructigena* was placed in and independent branch in the clade.

In 1983, Morgan-Jones introduced the new genus *Paraphoma* in order to accommodate phoma-like species with setose conidiomata [41]. However, the genus was later treated at section level within *Phoma* by Boerema [42]. De Gruyter [43] reinstated the genus, and placed it into the family *Phaeosphaeriaceae* based on a phylogenetic analysis. Twelve species are currently accepted in the genus (http://www.indexfungorum.org/names/Names.asp, accessed on 25 October 2021). *Paraphoma* spp. have been reported mainly as soil-borne phytopathogens, causing root and crown rot diseases [27,43,44]. *Paraphoma aquatica* differs from the other species of the genus because the ascomata lacks of ostiolar necks.

In 2012, Lawrey and Driederich introduced the new species *Phoma puncteliae*, isolated from the parasitized thalli of *Punctelia rudecta* [45]. Based on a phylogenetic analysis, Trakunyingcharoen and Crous [46] erected the new genus *Xenophoma* and placed it in the Phaeosphaeriaceae, designating *X. puncteliae* as its type species. The morphology of *Xenophoma* is similar to that of *Phoma*, differing by the production of cauliflower-shaped, uni- to multilocular conidiomata, and of the subspherical to ellipsoid conidia. Our new species, *X. microspora* differs from *X. puncteliae* (the phylogenetically nearest species) by the production of more than one ostiole per conidiomata and by the smaller bacilliform conidia.

The sexual stages of the freshwater ascomycetes have undergone a series of morphological adaptations to survive in aquatic environments. Many of them produce ascospores with appendages and/or mucilaginous sheaths, which facilitate their attachment to substrates into the water [47,48,49]. In this study, the sexual stage of *H. polypusiformis*, found on plant debris submerged in freshwater, produces ascospores with a mucilaginous sheath, a feature also found in other genera living in similar environments, such as *Murispora* and *Lolia* [48,50]. On the other hand, some coelomycetous fungi exclusively reported in freshwater habitats, such as *Aquasubmersa mircensis, Coelomyces aquaticus,* and *Lolia aquatica,* are characterized by the production of conidia with mucilaginous appendages [50,51], a feature not observed in our fungal strains, nor in the terricolous counterparts.

## Figures and Tables

**Figure 1 jof-07-01102-f001:**
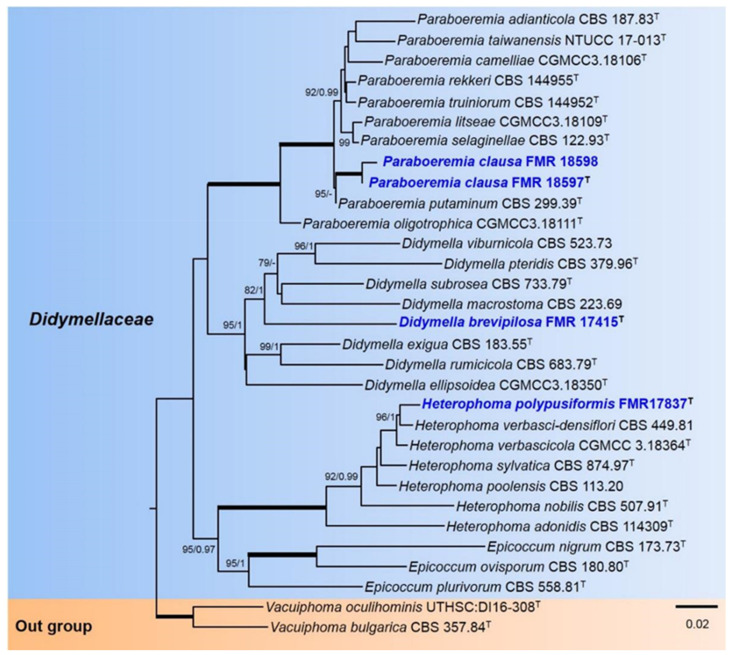
Phylogenetic tree inferred from a ML analysis based on a concatenated alignment of ITS, *rpb2,* and *tub2* sequences of 31 strains representing 31 species of *Didymellaceae*. The Bayesian posterior probabilities (PP) above 0.95 and the RAxML bootstrap support values (BS) above 70% are given at the nodes (PP/BS). Fully supported branches (1 PP/100 BS) are indicated in thicker lines. Newly proposed taxa are given in blue. Type strains are indicated by a superscript T. The tree was rooted with *Vacuiphoma bulgarica* CBS 357.84 and *V. oculihominis* UTHSC:DI16-308. Alignment length 1389 bp.

**Figure 2 jof-07-01102-f002:**
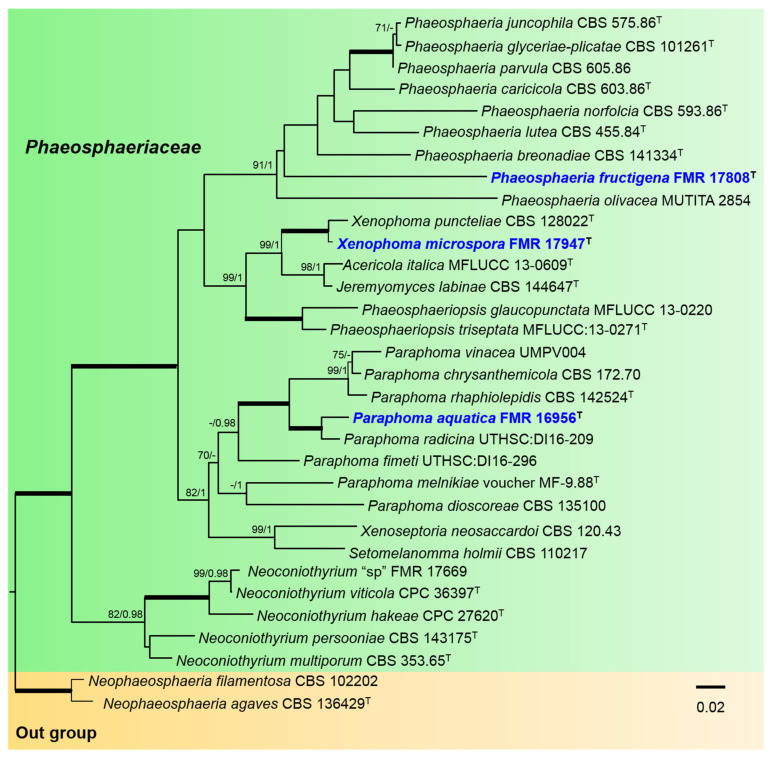
Phylogenetic tree inferred from a Maximum likelihood analysis based on a concatenated alignment of ITS, *rpb2, tub2,* and *tef-1* sequences of 32 strains representing 30 species of *Phaeosphaeriaceae*. The Bayesian posterior probabilities (PP) above 0.95 and the RAxML bootstrap support values (BS) above 70% are given at the nodes (PP/BS). Fully supported branches (1 PP/100 BS) are indicated thicker lines. Newly proposed taxa are given in blue. Type strains are indicated by a superscript T. The tree was rooted with *Neophaeosphaeria agaves* CBS 136429 and *N. filamentosa* CBS 102202. Alignment length 2256 bp.

**Figure 3 jof-07-01102-f003:**
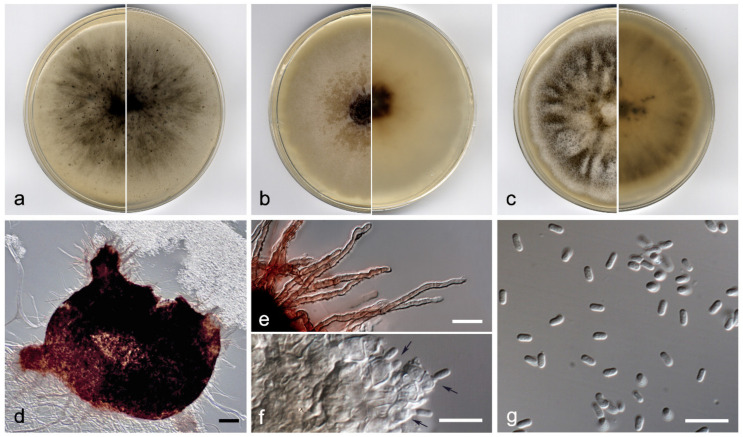
*Didymella brevipilosa* FMR 17415: (**a**) colonies on OA; (**b**) MEA; and (**c**) PDA after two weeks at 25 ± 1 °C (surface, left; reverse, right); (**d**) pycnidium; (**e**) setae; (**f**) conidiogenous cells (black arrows); (**g**) conidia. Scale bars: (**d**) = 50 µm, (**e**,**f**) = 10 µm.

**Figure 4 jof-07-01102-f004:**
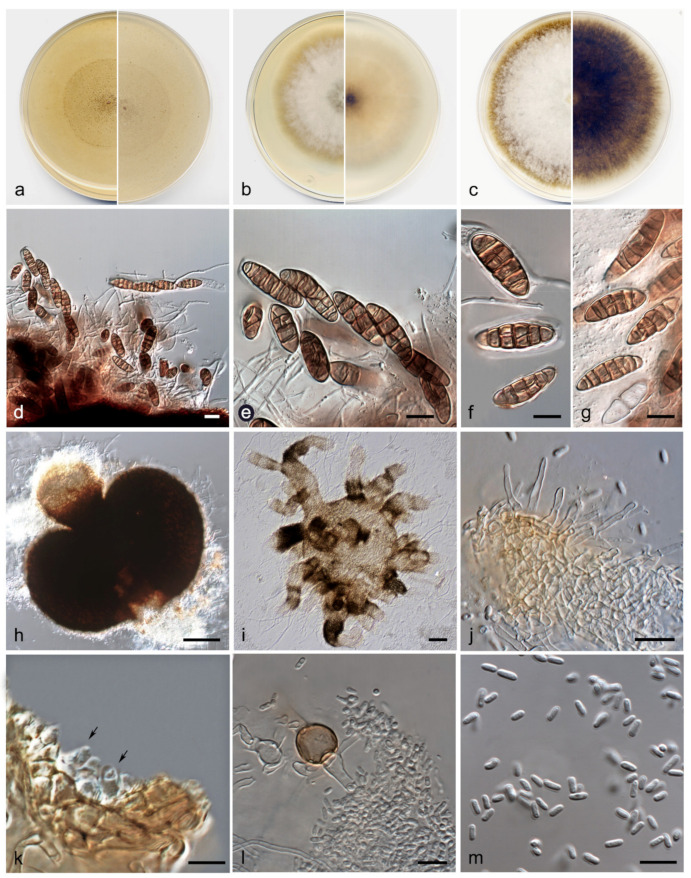
*Heterophoma polypusiformis* FMR 17837: (**a**) colonies on OA; (**b**) MEA; and (**c**) PDA after two weeks at 25 ± 1 °C (surface, left; reverse, right); (**d**,**e**) asci and ascospores; (**f**,**g**) ascospores (note the mucilaginous sheath in (**f**)); (**h**,**i**) pycnidium; (**j**) setae; (**k**) conidiogenous cells (black arrows); (**l**) chlamydospores and conidia; (**m**) conidia. Scale bars: (**h**,**i**) = 50 µm; (**d**–**g**), (**j**–**m**) = 10 µm.

**Figure 5 jof-07-01102-f005:**
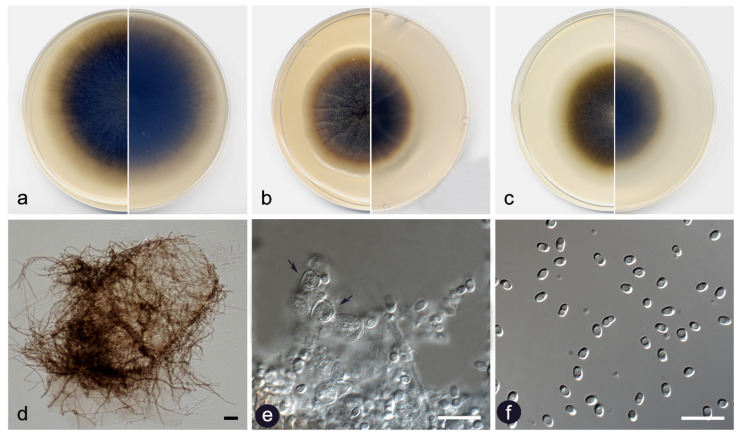
*Paraboeremia clausa* FMR 18597: (**a**) colonies on OA; (**b**) MEA; and (**c**) PDA after two weeks at 25 ± 1 °C (surface, left; reverse, right); (**d**) pycnidium; (**e**) conidiogenous cells (black narrows); (**f**) conidia. Scale bars: (**d**) = 50 µm, (**e**,**f**) = 10 µm.

**Figure 6 jof-07-01102-f006:**
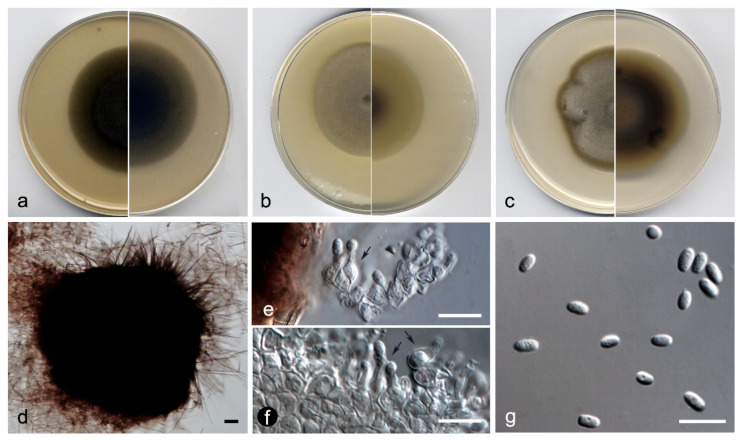
*Paraphoma aquatica* FMR 16956: (**a**) colonies on OA; (**b**) MEA; and (**c**) PDA after two weeks at 25 ± 1 °C (surface, left; reverse, right); (**d**) pycnidium; (**e**,**f**) conidiogenous cells (black arrows); (**g**) conidia. Scale bars: (**d**) = 25 µm, (**e**–**g**) = 10 µm.

**Figure 7 jof-07-01102-f007:**
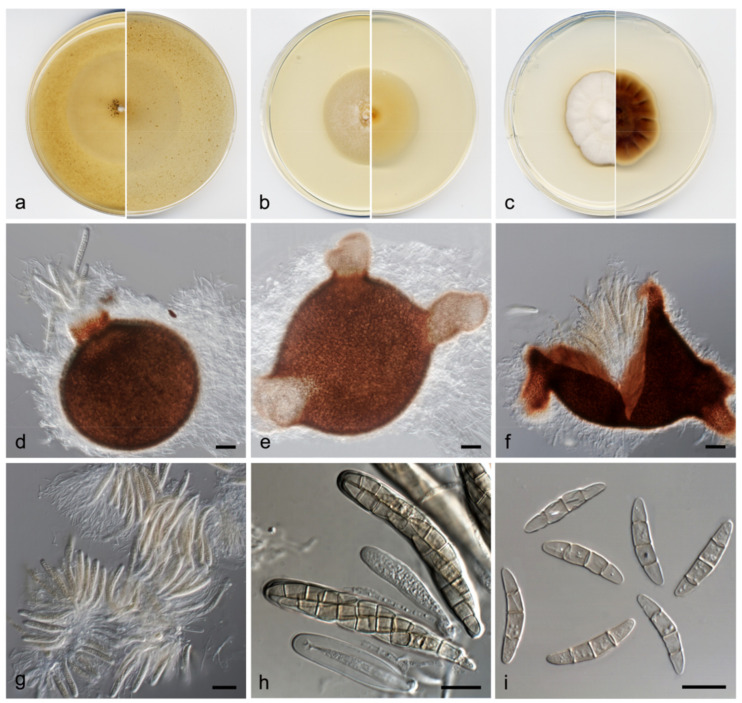
*Phaeosphaeria fructigena* FMR 17808: (**a**) colonies on OA; (**b**) MEA; and (**c**) PDA after two weeks at 25 ± 1 °C (surface, left; reverse, right); (**d**–**f**) ascomata ((**d**) expelling asci); (**g**,**h**) asci; (**i**) ascospores. Scale bars: (**d**–**f**) = 50 µm, (**g**–**i**) = 10 µm.

**Figure 8 jof-07-01102-f008:**
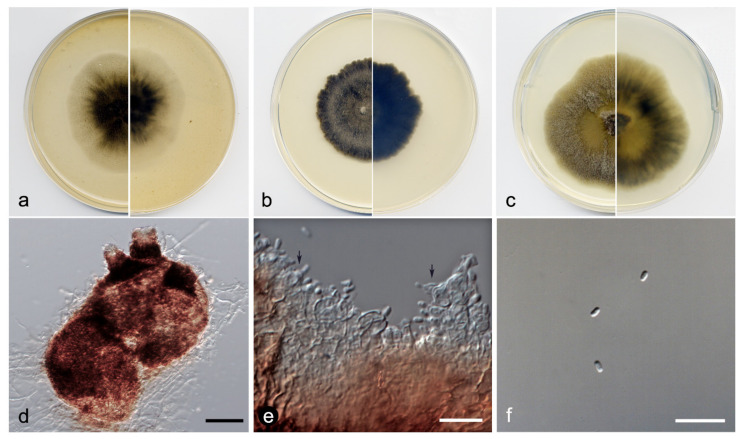
*Xenophoma microspora* FMR 17947: (**a**) colonies on OA; (**b**) MEA; and (**c**) PDA after two weeks at 25 ± 1 °C (surface, left; reverse, right); (**d**) pycnidium; (**e**) conidiogenous cells (black arrows); (**f**) conidia. Scale bars: (**d**) = 50 µm, (**e**,**f**) = 10 µm.

## Data Availability

Not applicable.

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
