# Peer review of "New Dothideomycetes from Freshwater Habitats in Spain"

_jof, 2021, doi:10.3390/jof7121102_

Round 1
Reviewer 1 Report
This work provides six new species of Dothideomycetes isolated from freshwater habitats in Spain, based on the multi-locus phylogeny and morphological characters. The paper has been well shaped. However, some issues should be addressed to improve it.
- The text is with some grammar and linguistic mistakes. It needs to be corrected throughout by a professional scientific writer and check again carefully.
- In the descriptions of new species in Didymellaceae, the authors need to make sure whether these strains produced setae, since there is not any species in this family reported with setae. And please provide more clear figures for this structure.
- For diagnosis, the shape of pycnidia is not a stable morphological character for species delimitation. If the morphological differences between close species are subtle, emphasize mainly their molecular differences in notes.
- Have the authors preserved any herbarium specimens of holotypes for the new species in public herbaria?
- Paraboeremia clausa: other “specimens examined” need to be provided. Since the two strains of this species have differences in sequences, have the authors check the morphological differences between them?
- In the discussion, we are more interested in the morphology, ecology, physiology or some special features of freshwater habitats, and the differences from species recovered from terrestrial environments.
An annotations text is provided.

Author Response
Review Report Form
Open Review
(x) I would not like to sign my review report
( ) I would like to sign my review report
English language and style
( ) Extensive editing of English language and style required
(x) Moderate English changes required
( ) English language and style are fine/minor spell check required
( ) I don't feel qualified to judge about the English language and style
Yes Can be improved Must be improved Not applicable
Does the introduction provide sufficient background and include all relevant references?
(x) ( ) ( ) ( )
Is the research design appropriate?
(x) ( ) ( ) ( )
Are the methods adequately described?
(x) ( ) ( ) ( )
Are the results clearly presented?
(x) ( ) ( ) ( )
Are the conclusions supported by the results?
(x) ( ) ( ) ( )
Comments and Suggestions for Authors
This work provides six new species of Dothideomycetes isolated from freshwater habitats in Spain, based on the multi-locus phylogeny and morphological characters. The paper has been well shaped. However, some issues should be addressed to improve it.
The text is with some grammar and linguistic mistakes. It needs to be corrected throughout by a professional scientific writer and check again carefully.
RESPONSE: The article has been reviewed by a native English professor with a large trajectory in correction of scientific articles.
In the descriptions of new species in Didymellaceae, the authors need to make sure whether these strains produced setae, since there is not any species in this family reported with setae. And please provide more clear figures for this structure.
RESPONSE: The presence of pycnidial setae in Didymella brevipilosa is now clearly illustrated in the corresponding plate (Figure E). Regarding the other species we described with setae, Heterophoma polypusiformis, such structures are illustrated in the Figure J. On the other hand, the pycnidia of Paraboeremia clausa are only covered by hyphae, which was described in the original version of the manuscript.
For diagnosis, the shape of pycnidia is not a stable morphological character for species delimitation. If the morphological differences between close species are subtle, emphasize mainly their molecular differences in notes.
RESPONSE: Molecular differences in base pairs have been introduced as Notes.
Have the authors preserved any herbarium specimens of holotypes for the new species in public herbaria?
RESPONSE: The type specimens and the ex-type strains of the novel fungi have been deposited at the Westerdijk Fungal Biodiversity Institute culture collection of fungi and yeasts (CBS, Utrecht, The Netherlands). The CBS numbers have been added in the new version of the manuscript.
Paraboeremia clausa: other “specimens examined” need to be provided. Since the two strains of this species have differences in sequences, have the authors check the morphological differences between them?
RESPONSE: Done.
In the discussion, we are more interested in the morphology, ecology, physiology or some special features of freshwater habitats, and the differences from species recovered from terrestrial environments.
RESPONSE: The Discussion has been re-edited according to the suggestions of the reviewer.
Reviewer 2 Report
Comments on manuscript:
New Dothideomycetes from freshwater habitats in Spain 2
Viridiana Magaña-Dueñas, José Francisco Cano-Lira, Alberto Mi-guel Stchigel
This is an excellent manuscript introducing a number of Phoma-like taxa, a group that is often dismissed because of the few taxonomical characters for their identification. The manuscript is well written (with few exceptions highlighted in the retuned manuscript), supported by clear figures which illustrate the features of the different taxa and supported by phylogenetic data.
Issues to address:
The diagnoses/notes after the discerptions are short and not as useful as they might be. In many cases the fact that pycnidia produce more then one ostiole is used as a diagnostic character. This is not a reliable character to use, and in my opinion is artefact of growth in culture. No mention is made of any reproductive structures on the collected debris.
Words highlighted should be looked at and clarified.
Line 31: might be useful to refer to the more up to date website: https://www.freshwaterfungi.org
Line 54: It is a pity that plant debris collected submerged in freshwater is not illustrated or dimensions of the material indicated. This could be useful for others working on aquatic fungi.
Line 128: branches?
Line 163: In the legends the word pycnidia is used, whereas only a pycnidium is illustrated.
Line 215-215: Legend Ascospores, Black arrows.
Line 257-260: This sentence does not red well, rephrase.
Line 337: delete the words: that produces no need to say this.
Line 406: No reason given why LSU data was not used ion your study,
Line 460: the latter part of the sentence is not clear.
A few points spotted in the refer fences. Low case should be use in titles of manuscripts, unless the titles of books etc.
Recommendation minor revision.

Author Response
Review Report Form
Open Review
(x) I would not like to sign my review report
( ) I would like to sign my review report
English language and style
( ) Extensive editing of English language and style required
( ) Moderate English changes required
(x) English language and style are fine/minor spell check required
( ) I don't feel qualified to judge about the English language and style
Yes Can be improved Must be improved Not applicable
Does the introduction provide sufficient background and include all relevant references?
(x) ( ) ( ) ( )
Is the research design appropriate?
(x) ( ) ( ) ( )
Are the methods adequately described?
(x) ( ) ( ) ( )
Are the results clearly presented?
(x) ( ) ( ) ( )
Are the conclusions supported by the results?
(x) ( ) ( ) ( )
Comments and Suggestions for Authors
Comments on manuscript:
New Dothideomycetes from freshwater habitats in Spain 2
Viridiana Magaña-Dueñas, José Francisco Cano-Lira, Alberto Miguel Stchigel
This is an excellent manuscript introducing a number of Phoma-like taxa, a group that is often dismissed because of the few taxonomical characters for their identification. The manuscript is well written (with few exceptions highlighted in the retuned manuscript), supported by clear figures which illustrate the features of the different taxa and supported by phylogenetic data.
Issues to address:
The diagnoses/notes after the discerptions are short and not as useful as they might be. In many cases the fact that pycnidia produce more then one ostiole is used as a diagnostic character. This is not a reliable character to use, and in my opinion is artefact of growth in culture. No mention is made of any reproductive structures on the collected debris.
RESPONSE: We agree with the reviewer in that the morphology of the pycnidia, especially when the fungi grown in pure culture, is not a robust taxonomic character to take usually into account. However, the observation of numerous necks in Heterophoma polypusiformis, despite these are produced in “artificial” substrata, is a very unusual and remarkable (and reproducible) character, which also is present in H. verbasci-densiflori, its phylogenetically closest species. About the finding of fruiting bodies on plant debris, this information has been included in the manuscript, as well as the bp differences among our new species and the phylogenetically closest ones.
Words highlighted should be looked at and clarified.
RESPONSE: Done.
Line 31: might be useful to refer to the more up to date website: https://www.freshwaterfungi.org
RESPONSE: We greatly appreciate the suggestion to use such database information; in fact, this is one of the websites highly consulted by us. However, we found more updated information regarding the freshwater ascomycetes (including those previously reported by us) in http://fungi.life.illinois.edu/.
Line 54: It is a pity that plant debris collected submerged in freshwater is not illustrated or dimensions of the material indicated. This could be useful for others working on aquatic fungi.
RESPONSE: The plant material collected from freshwater environments was mostly highly degraded, making it impossible to reach its botanical identification. For further works, the suggestion of the reviewer to document better the original material (including pictures of them) will be taken into account.
Line 128: branches?
RESPONSE: Done.
Line 163: In the legends the word pycnidia is used, whereas only a pycnidium is illustrated.
RESPONSE: Done.
Line 215-215: Legend Ascospores, Black arrows.
RESPONSE: Done.
Line 257-260: This sentence does not red well, rephrase.
RESPONSE: Done.
Line 337: delete the words: that produces no need to say this.
RESPONSE: Done.
Line 406: No reason given why LSU data was not used ion your study,
RESPONSE: For the Phaeosphaeriaceae, a phylogenetic tree was made using the LSU marker, which can be seen in Figure S1. For the Didymellaceae, we did not included the LSU phylogenetic tree because it reported a very poor discrimination among the taxa, including genera.
Line 460: the latter part of the sentence is not clear.
RESPONSE: Modified.
A few points spotted in the refer fences. Low case should be use in titles of manuscripts, unless the titles of books etc.
RESPONSE: Done.